# Boron Trifluoride Etherate Promoted Regioselective 3-Acylation of Indoles with Anhydrides

**DOI:** 10.3390/molecules27238281

**Published:** 2022-11-28

**Authors:** Yunyun Zheng, Jiuling Li, Kai Wei

**Affiliations:** Henan Engineering Research Center of Funiu Mountain’s Medical Resources Utilization and Molecular Medicine, School of Medical Sciences, Pingdingshan University, Pingdingshan 467000, China

**Keywords:** indoles, acylation, boron trifluoride etherate, anhydrides

## Abstract

An efficient, high-yielding and scalable procedure for the regioselective 3-acylation of indoles with anhydrides promoted by boron trifluoride etherate under mild conditions was reported. This novel protocol provided a simple way to prepare 3-(benzofuran-2-yl) indole in three steps.

## 1. Introduction

The indole and its derivatives are versatile structural motifs in organic and medicinal chemistry [1,2,3,4,5,6,7,8,9], which have been regarded as “privileged fragments” in biologically active natural products and pharmaceutical compounds. Among them, 3-acylindole moiety has not only been found to act as a proven pharmacophore element in bioactive molecules such as Analogue, Bruceolline and so on [10,11,12,13,14,15,16,17,18,19] (Figure 1), but has also served as a versatile intermediate in the synthesis of indole derivatives [20,21,22,23,24,25,26,27,28,29,30]. For examples, an analogue of deoxytopsentin displayed potent low nanomolar inhibitory activity against MRSA PK with concomitant significant selectivity for MRSA PK over human PK orthologues. Computational studies suggest that these potent MRSA PK inhibitors occupy a region of the small interface of the enzyme tetramer where amino acid sequence divergence from common human PK orthologues may contribute to the observed selectivity [14]. Oxi8006 is one of the first indole-based, colchicine-site-binding inhibitors of tubulin assembly into microtubules. OXi8006 is a potent inhibitor of tubulin polymerization (IC_50_ = 1.1 μM) and competes with radiolabeled colchicine at the colchicine binding site of tubulin. OXi8006 was shown to be cytotoxic against three evaluated human cancer cell lines, NCI-H460, DU-145 and SK-OV-3, with an average GI50 of 25.7 nM [18,19]. Consequently, developing a more efficient and practical protocol for the synthesis of 3-acylindoles has gained considerable attention [31,32,33,34,35,36,37,38,39,40,41,42,43,44,45,46].

Regioselective functionalization of indoles is one of the most important challenges in the field of indole chemistry, especially in the acylation of free (N-H) indoles. Though many strategies, such as the metal-catalyzed intramolecular oxidative coupling reaction for the preparation of 3-acylindoles, have been developed in recent decades [31,32,33,34,35,36,37], the requirement for expensive and complex ligands and mental catalysts in this novel protocol make few of them suitable for lab or industrial preparation today. The Friedel–Crafts reaction was still regarded as the most promising, practical and convenient protocol [38,39,40,41,42]. AlCl3 [38,39,40], SnCl4 [40], TiCl4 [40], ZrCl4 [41] dialkylaluminum chloride [43,44,45,46] were the most commonly used reagents to promote acylation due to their easy availability and high reactivity. However, these common Lewis acids suffered from some limitations. Some of them required additional protection and deprotection steps to eliminate 1-acylation and 1,3-diacylation [41]. In the above developed methods, most of the reagents were poor-moisture-tolerant and air-sensitive, and the presence of a metal ion resulted in a laborious and frustrated workup. In addition, most of the metal ions are toxic and must be carefully removed from the products, especially for the drug and pharmaceutical industry. Additionally, environmental awareness has also made them not preferred in this transformation, especially in the large-scale processes. The application of more environmentally benign solid Lewis acids or Brønsted acidic ionic liquids such as modified zeolites or bisulfate in this acylation process have also been reported [47,48], but only very limited substrates have been used, and the preparation of the Brønsted acidic ionic liquid was more complex than the commercial reagents. Accordingly, a metal-free, more environmentally benign regioselective acylation procedure to prepare 3-acylindoles under mild reaction conditions and simple workup is highly desirable. 

Herein, we report a high regioselective and scalable protocol (Figure 1) for the 3-acylatation of indoles with anhydrides in the presence of boron trifluoride etherate, a very common and easy-to-handle Lewis acid that has been widely used in organic reactions [49].

## 2. Results and Discussion

At the beginning of our studies, acylation of indole **1a** with acetic anhydride **2a** in different solvents in the presence of BF_3_·Et_2_O was explored (Table 1, entries 1–6). We found that the acylation reaction could occur in DCM, DCE, CHCl_3_, MeCN or 1,4-dioxane, and that DCM gave the best results (Table 1, entry 1). Then, the amount of BF_3_·Et_2_O (Table 1, entries 7–10) and anhydride (Table 1, entries 11–13) were investigated, which revealed that when the ratio of indole, anhydride and BF_3_·Et_2_O was 1:1.2:1, the yield of 3-acylindole **3aa** achieved 83% (Table 1, entry 12). Another important point is that in the absence of BF_3_·Et_2_O (entry 7), no desired product was achieved. Further screening of the reaction temperature showed that room temperature was the best choice (Table 1, entries 14–15).

With the optimal reaction conditions, the aliphatic, alicyclic and aryl anhydrides were subjected to investigate the scope of anhydrides in the acylation reaction. The results are summarized in Table 2. The aliphatic, alicyclic and aryl anhydrides could react with indole **1a** smoothly to furnish the desired products **3** in good-to-excellent yields (Table 2, entries 1–6). However, no desired products were observed for 4-chloro and 4-nitro benzoic anhydrides, perhaps due to the solubility of anhydrides in DCM (Table 2, entries 7–8).

To test the scope of the present protocols, various substituents at different positions of indole ring, including the 1- and 2-substitued indoles, were investigated. As shown in Table 3, both electron-donating and electron-withdrawing substituents in indoles gave the corresponding 3-acylindoles in good-to-excellent yields (from 53% to 93%). The position of the substituents and the electronic nature on the indole ring did not play important roles; only the indoles with electron-withdrawing groups afforded a slightly better yield. The 1-methylindole (**1b** entry1–4) and 2-phenyl-1*H*-indole (**1d** entry9–12) needed a longer time to finish the reaction. From Table 3, we can see the aliphatic anhydrides usually gave higher yields than the aryl ones. Furthermore, the structure of the 3-acylation products **3ec** was further confirmed by X-ray diffraction analysis (see Appendix A).

Compared with the aforementioned protocols promoted by the common Lewis acids or dialkylaluminum chloride, the more moisture-tolerant, air-stable and easy-to-handle BF_3_·Et_2_O provided an efficient entry to 3-acylindoles. It is also worth noting that when the acylation of indole **1a** with acetic anhydride was carried out on more than a 10 g (0.1 mol, 11.7 g) scale, the 3-acylation reaction still provided 80% yield (Figure 2), which would lead these compounds to be applied more easily.

3-Benzofuranyl indoles are one of the most important scaffolds in heterocycle chemistry [50,51,52]. The popular methods to synthesize these compounds are the transition-metal catalyzed cross-coupling reaction, which suffered from limited substrates, requirement of complex ligands, and trace amounts of metals in the products [50,51], and preparation via the Sc(OTf)3-mediated Meinwald epoxide rearrangement of benzofuran-2-yl oxirane with aryl-hydrazine [52]. During the studies of O-arylation and [3,3]-rearrangement with diaryliodonium salts in our group [53,54], we surmised that 3-benzofuran-2-yl indole could be synthesized easily from **3aa** (Figure 3). Firstly, **3aa** was converted to the oxime **4**, which further underwent a C-O formation with diphenyliodonium triflate to give the O-phenyl-oxime **5** which could undergo a [3,3]-rearrangement in acid condition [55] to accomplish the 3-benzofuranyl indoles **6** with 67% yield.

## 3. Experimental Section

Unless otherwise noted, all reactions were performed under air atmosphere, and commercial materials and solvents were used directly without further purification. All reagents were weighed and handled in air at room temperature. ^1^H-NMR and ^13^C-NMR spectra were recorded on Bruker Avance 400 and 600 spectrometers. Chemical shifts are reported in parts per million (δ) referenced to tetramethylsilane (0.0 ppm), chloroform (7.26 ppm or 77.0 ppm) and DMSO (2.5 ppm or 39.5 ppm), respectively. Data for ^1^H-NMR and ^13^C-NMR spectroscopy are reported as follows: chemical shift (δ ppm), multiplicity (s = singlet, d = doublet, t = triplet, q = quartet, m = multiplet, br = broad), coupling constant (Hz), integration. X-ray single crystal diffraction data were recorded on Bruker D8 QUEST and Bruker APEX DUO. High Resolution Mass spectra were taken on an AB QSTAR Pulsar mass spectrometer or Aglient LC/MSD TOF mass spectrometer. Melting points were measured on a Hanon MP 430 auto melting-point system and are uncorrected. Silica gel (200–300 mesh) for column chromatography and silica GF254 for TLC were obtained from Merck Chemicals Co. Ltd. (Shanghai, China). Petroleum ether with the boiling range of 60–90 °C was used for column chromatography. All reactions were conducted in dried glassware under a positive pressure of dry nitrogen or argon. Reagents and starting materials were accordingly transferred via syringe or cannula. Reaction temperatures refer to the external oil bath temperature and are uncorrected. Conditional optimization and copies of spectroscopic characterization of all new compounds are available in the Appendix A.

### 3.1. General Procedure for the Synthesis of Product (***3aa*–*3jd***)

A mixture of indole **1** (0.5 mmol), anhydride **2** (0.6 mmol), and BF_3_·Et_2_O (BF_3_ 46.5%) (0.5 mmol, 64 μL) in DCM was stirred at room temperature for the desired time. After the reaction was completed, saturated sodium bicarbonate (10 mL) was added. The reaction mixture was stirred for 5 min and then extracted with ethyl acetate (3 × 10 mL). The organic layer was dried over anhydrous Na_2_SO_4_ and, after evaporation of the solvent under reduced pressure, the residue was purified by column chromatography on silica gel using petroleum ether/EtOAc (6:1 to 2:1) as the eluents to give product **3**.

*1-(1H-Indol-3-yl)ethanone (***3aa***)*. Pale yellow solid; 66 mg, 83% yield, m.p. 195–196 °C (lit. 192–193 °C); [41] ^1^H NMR (500 MHz, DMSO-*d*_6_) δ = 11.94 (s, 1H), 8.31 (s, 1H), 8.21–8.14 (m, 1H), 7.50–7.42 (m, 1H), 7.23–7.15 (m, 2H), 2.45 (s, 3H); ^13^C NMR (125 MHz, DMSO-*d*_6_) δ = 193.1, 137.1, 134.7, 125.7, 123.1, 122.1, 121.82, 117.3, 112.5, 27.7. HRMS (ESI): *m*/*z* [M+H]^+^ calcd for C_10_H_10_NO: 160.0757; found: 160.0761.

*1-(1H-Indol-3-yl)propan-1-one (***3ab***)*. White solid; 79 mg, 91% yield, m.p. 161–162 °C (lit. 162–163 °C) [39]; ^1^H NMR (500 MHz, DMSO-*d*_6_) δ = 11.91 (s, 1H), 8.31 (d, *J* = 3.1 Hz, 1H), 8.20 (d, *J* = 7.2 Hz, 1H), 7.46 (d, *J* = 7.4 Hz, 1H), 7.23–7.13 (m, 2H), 2.88 (q, *J* = 7.4 Hz, 2H), 1.11 (t, *J* = 7.4 Hz, 3H); ^13^C NMR (125 MHz, DMSO-*d*_6_) *δ* = 196.3, 137.1, 133.9, 125.9, 123.1, 122.0, 121.8, 116.5, 112.5, 32.3, 9.6. HRMS (ESI): *m*/*z* [M+H]^+^ calcd for C_11_H_12_NO: 174.0913; found: 174.0915.

*1-(1H-Indol-3-yl)butan-1-one (***3ac***)*. White solid; 80 mg, 85% yield, m.p. 176–177 °C (lit. 181–182 °C) [56]; ^1^H NMR (500 MHz, DMSO-*d*_6_) δ = 11.91 (s, 1H), 8.33 (d, *J* = 3.1 Hz, 1H), 8.20 (d, *J* = 7.1 Hz, 1H), 7.51–7.43 (m, 1H), 7.22–7.15 (m, 2H), 2.82 (t, *J* = 7.3 Hz, 2H), 1.71–1.63 (m, 2H), 0.94 (t, *J* = 7.4 Hz, 3H); ^13^C NMR (125 MHz, DMSO-*d*_6_) δ = 195.8, 137.1, 134.2, 125.9, 123.1, 122.0, 121.9, 117.0, 112.5, 41.2, 18.8, 14.4. HRMS (ESI): *m*/*z* [M+H]^+^ calcd for C_12_H_14_NO: 188.1070; found: 188.1068.

*Cyclohexyl(1H-indol-3-yl)methanone (***3ad***)*. White solid; 82 mg, 72% yield, m.p. 194–195 °C; ^1^H NMR (500 MHz, DMSO-*d*_6_) δ = 11.92 (s, 1H), 8.36 (d, *J* = 3.1 Hz, 1H), 8.22 (d, *J* = 7.2 Hz, 1H), 7.48–7.46 (m, 1H), 7.22–7.16 (m, 2H), 3.21–3.16 (m, 1H), 1.80–1.76 (m, 4H), 1.69 (d, *J* = 12.8 Hz, 1H), 1.50–1.37 (m, 4H), 1.25–1.16 (m, 1H); ^13^C NMR (125 MHz, DMSO-*d*_6_) δ = 199.3, 137.3, 133.9, 126.2, 123.1, 122.0, 121.9, 115.6, 112.5, 46.6, 30.2, 26.2, 25.8. HRMS (ESI): *m*/*z* [M+H]^+^ calcd for C_15_H_18_NO: 228.1383; found: 228.1385.

*(1H-Indol-3-yl)(phenyl)methanone (***3ae***)*. Pale yellow solid; 84 mg, 76% yield, m.p. 223–224 °C (lit. 243–245 °C) [41]; ^1^H NMR (500 MHz, DMSO-*d*_6_) δ = 12.10 (s, 1H), 8.26 (d, *J* = 7.3 Hz, 1H), 7.95 (d, *J* = 3.1 Hz, 1H), 7.80–7.70 (m, 2H), 7.63–7.60 (m, 1H), 7.57–7.52 (m, 3H), 7.29–7.22 (m, 2H); ^13^C NMR (125 MHz, DMSO-*d*_6_) δ = 190.5, 140.9, 137.2, 136.3, 131.6, 128.9, 128.8, 126.7, 123.6, 122.4, 121.9, 115.5, 112.7. HRMS (ESI): *m*/*z* [M+H]^+^ calcd for C_15_H_12_NO: 222.0913; found: 222.0911.

*(1H-Indol-3-yl)(4-methoxyphenyl)methanone (***3af***)*. Pale yellow solid; 100 mg, 80% yield, m.p. 205–206 °C (lit. 208 °C) [57]; ^1^H NMR (500 MHz, DMSO-*d*_6_) δ = 12.01 (s, 1H), 8.23 (d, *J* = 7.0 Hz, 1H), 7.95 (d, *J* = 3.1 Hz, 1H), 7.83–7.80 (m, 2H), 7.53–7.51 (m, 1H), 7.26–7.21 (m, 2H), 7.09–7.06 (m, 2H), 3.86 (s, 3H); ^13^C NMR (125 MHz, DMSO-*d*_6_) δ = 189.2, 162.2, 137.1, 135.3, 133.45, 131.1, 126.9, 123.4, 122.1, 121.9, 115.6, 114.1, 112.6, 55.9. HRMS (ESI): *m*/*z* [M+H]^+^ calcd for C_16_H_14_NO_2_: 252.1019; found: 252.1018.

*(1H-Indol-3-yl)(m-tolyl)methanone (***3ag***)*. Pale yellow solid; 83 mg, 71% yield, m.p. 234–236 °C; ^1^H NMR (500 MHz, DMSO-*d*_6_) δ = 12.05 (s, 1H), 8.27 (d, *J* = 7.1 Hz, 1H), 7.93 (d, *J* = 2.8 Hz, 1H), 7.62–7.57 (m, 2H), 7.54 (d, *J* = 7.4 Hz, 1H), 7.44–7.40 (m, 2H), 7.29–7.22 (m, 2H), 2.41 (s, 3H). ^13^C NMR (125 MHz, DMSO-*d*_6_) δ = 190.6, 141.1, 138.2, 137.2, 136.1, 132.1, 129.3, 128.7, 126.7, 126.1, 123.6, 122.3, 122.0, 115.6, 112.7, 21.5. HRMS (ESI): *m*/*z* [M+H]^+^ calcd for C_16_H_14_NO: 236.1070; found: 236.1071.

*1-(1-Methyl-1H-indol-3-yl)ethanone (***3ba***)*. White solid; 75 mg, 87% yield, m.p. 102–103 °C (lit. 105–107 °C) [58]; ^1^H NMR (500 MHz, DMSO-*d*_6_) δ = 8.32 (s, 1H), 8.20 (d, *J* = 7.7 Hz, 1H), 7.53 (d, *J* = 8.1 Hz, 1H), 7.30–7.26 (m, 1H), 7.25–7.21 (m, 1H), 3.86 (s, 3H), 2.43 (s, 3H); ^13^C NMR (125 MHz, DMSO-*d*_6_) δ = 192.5, 138.4, 138.4, 137.7, 126.2, 123.2, 122.4, 121.9, 116.1, 110.9, 33.5, 27.7. HRMS (ESI): *m*/*z* [M+H]^+^ calcd for C_11_H_12_NO: 174.0913; found: 174.0919.

*1-(1-Methyl-1H-indol-3-yl)propan-1-one (***3bb***)*. White solid; 77 mg, 82% yield, m.p. 72–73 °C; ^1^H NMR (500 MHz, DMSO-*d*_6_) δ = 8.33 (s, 1H), 8.22 (d, *J* = 7.8 Hz, 1H), 7.53 (d, *J* = 8.0 Hz, 1H), 7.30–7.25 (m, 1H), 7.25–7.21 (m, 1H), 3.86 (s, 3H), 2.84 (q, *J* = 7.4 Hz, 2H), 1.12 (t, *J* = 7.4 Hz, 3H); ^13^C NMR (125 MHz, DMSO-*d*_6_) δ = 195.7, 137.7, 137.6, 126.3, 123.1, 122.4, 121.9, 115.3, 110.9, 33.5, 32.4, 9.6. HRMS (ESI): *m*/*z* [M+H]^+^ calcd for C_12_H_14_NO: 188.1070; found: 188.1073.

*1-(1-Methyl-1H-indol-3-yl)butan-1-one (***3bc***)*. Brown oil; 71 mg, 71% yield; ^1^H NMR (500 MHz, DMSO-*d*_6_) δ = 8.30 (d, *J* = 2.4 Hz, 1H), 8.28–8.24 (m, 1H), 7.50 (d, *J* = 7.7 Hz, 1H), 7.29–7.20 (m, 2H), 3.84 (d, *J* = 2.6 Hz, 3H), 2.80–2.75 (m, 2H), 1.72–1.64 (m, 2H), 0.97–0.91 (m, 3H); ^13^C NMR (125 MHz, DMSO-*d*_6_) δ = 195.5, 137.9, 137.7, 126.3, 123.2, 122.4, 122.0, 115.8, 110.9, 41.3, 33.5, 18.8, 14.3. HRMS (ESI): *m*/*z* [M+H]^+^ calcd for C_13_H_16_NO: 202.1226; found: 202.1228.

*(1-Methyl-1H-indol-3-yl)(phenyl)methanone (***3bd***)*. White solid; 62 mg, 53% yield, m.p. 110–111 °C (lit. 116–118 °C) [41]; ^1^H NMR (500 MHz, DMSO-*d*_6_) δ = 8.29 (d, *J* = 7.5 Hz, 1H), 8.02 (s, 1H), 7.80 (d, *J* = 1.0 Hz, 1H), 7.79–7.78 (m, 1H), 7.64–7.56 (m, 3H), 7.56–7.53 (m, 1H), 7.36–7.33 (m, 1H), 7.32–7.28 (m, 1H), 3.89 (s, 3H); ^13^C NMR (125 MHz, DMSO-*d*_6_) δ = 189.9, 141.0, 139.9, 137.8, 131.5, 128.9, 128.8, 127.1, 123.7, 122.8, 122.1, 114.3, 111.1, 33.6. HRMS (ESI): *m*/*z* [M+H]^+^ calcd for C_16_H_14_NO: 236.1070; found: 236.1074.

*1-(2-Methyl-1H-indol-3-yl)ethanone (***3ca***)*. White solid; 59 mg, 68% yield, m.p. 203–204 °C (lit. 201–202 °C) [59]; ^1^H NMR (500 MHz, DMSO-*d*_6_) δ = 11.84 (s, 1H), 8.04–7.99 (m, 1H), 7.39–7.34 (m, 1H), 7.16–7.11 (m, 2H), 2.68 (s, 3H), 2.51 (s, 3H); ^13^C NMR (125 MHz, DMSO-*d*_6_) δ = 193.4, 144.6, 135.1, 127.4, 122.2, 121.7, 121.0, 113.9, 111.6, 31.4, 15.4. HRMS (ESI): *m*/*z* [M+H]^+^ calcd for C_11_H_12_NO: 174.0913; found: 174.0915.

*1-(2-Methyl-1H-indol-3-yl)propan-1-one (***3cb***)*. Yellow solid; 66 mg, 71% yield, m.p. 140–141 °C (lit. 150 °C) [60]; ^1^H NMR (500 MHz, DMSO-*d*_6_) δ = 11.82 (s, 1H), 8.03–7.99 (m, 1H), 7.39–7.35 (m, 1H), 7.16–7.11 (m, 2H), 2.90 (q, *J* = 7.2 Hz, 2H), 2.68 (s, 3H), 1.11 (t, *J* = 7.2 Hz, 3H); ^13^C NMR (125 MHz, DMSO-*d*_6_) δ = 196.5, 144.3, 135.2, 127.2, 122.1, 121.7, 121.1, 113.4, 111.6, 35.4, 15.6, 8.8. HRMS (ESI): *m*/*z* [M+H]^+^ calcd for C_12_H_14_NO: 188.7070; found: 188.1076.

*1-(2-Methyl-1H-indol-3-yl)butan-1-one (***3cc***)*. White solid; 75 mg, 74% yield, m.p. 143–144 °C (lit. 157–158 °C) [60]; ^1^H NMR (500 MHz, DMSO-*d*_6_) δ = 11.82 (s, 1H), 8.02–7.97 (m, 1H), 7.39–7.34 (m, 1H), 7.17–7.09 (m, 2H), 2.86 (t, *J* = 7.2 Hz, 2H), 2.68 (s, 3H), 1.67 (h, *J* = 7.3 Hz, 2H), 0.97 (t, *J* = 7.4 Hz, 3H); ^13^C NMR (125 MHz, DMSO-*d*_6_) δ = 195.9, 144.3, 135.2, 127.1, 122.1, 121.7, 121.0, 113.6, 111.6, 44.3, 17.7, 15.6, 14.4. HRMS (ESI): *m*/*z* [M+H]^+^ calcd for C_13_H_16_NO: 202.1226; found: 202.1232.

*(2-Methyl-1H-indol-3-yl)(phenyl)methanone (***3cd***)*. Pale yellow solid; 66,mg, 56% yield, m.p. 180–181 °C (lit. 183–185 °C) [57]; ^1^H NMR (500 MHz, DMSO-*d*_6_) δ = 11.96 (s, 1H), 7.61–7.57 (m, 3H), 7.53–7.48 (m, 2H), 7.38 (d, *J* = 8.0 Hz, 1H), 7.32 (d, *J* = 8.0 Hz, 1H), 7.15–7.09 (m, 1H), 7.04–6.98 (m, 1H), 2.38 (s, 3H); ^13^C NMR (125 MHz, DMSO-*d*_6_) δ = 192.2, 145.0, 142.1, 135.4, 131.5, 128.8, 128.5, 127.7, 122.3, 121.4, 120.5, 112.9, 111.7, 14.7. HRMS (ESI): *m*/*z* [M+H]^+^ calcd for C_16_H_14_NO: 236.1070; found: 236.1077.

*1-(2-Phenyl-1H-indol-3-yl)ethanone (***3da***)*. White solid; 74 mg, 62% yield, m.p. 231–232 °C (lit. 220–222 °C) [61]; ^1^H NMR (500 MHz, DMSO-*d*_6_) δ = 12.11 (s, 1H), 8.20 (d, *J* = 7.5 Hz, 1H), 7.65 (d, *J* = 3.6 Hz, 2H), 7.56 (d, *J* = 3.6 Hz, 3H), 7.42 (d, *J* = 7.7 Hz, 1H), 7.25–7.18 (m, 2H), 2.07 (s, 3H); ^13^CNMR (125 MHz, DMSO-*d*_6_)δ = 194.0, 145.4, 135.9, 133.2, 130.5, 129.8, 128.9, 127.5, 123.3, 122.20, 122.0, 114.7, 112.0, 30.6. HRMS (ESI): *m*/*z* [M+H]^+^ calcd for C_16_H_14_NO: 236.1070; found: 236.1068.

*1-(2-Phenyl-1H-indol-3-yl)propan-1-one (***3db***)*. White solid; 80 mg, 65% yield, m.p. 192–193 °C (lit. 189 °C) [62]; ^1^H NMR (500 MHz, DMSO-*d*_6_) δ = 12.07 (s, 1H), 8.18 (d, *J* = 7.2 Hz, 1H), 7.65–7.61 (m, 2H), 7.59–7.54 (m, 3H), 7.42 (dd, *J* = 7.1, 1.2 Hz, 1H), 7.25–7.17 (m, 2H), 2.42 (q, *J* = 7.3 Hz, 2H), 0.92 (t, *J* = 7.3 Hz, 3H); ^13^C NMR (125 MHz, DMSO-*d*_6_) δ = 197.5, 144.6, 135.9, 133.4, 130.28, 129.7, 128.9, 127.5, 123.2, 122.1, 122.0, 114.2, 112.0, 34.8, 9.3. HRMS (ESI): *m*/*z* [M+H]^+^ calcd for C_17_H_16_NO: 250.1226; found: 250.1232.

*1-(2-Phenyl-1H-indol-3-yl)butan-1-one (***3dc***)*. White solid; 92 mg, 70% yield, m.p. 168–169 °C; ^1^H NMR (500 MHz, DMSO-*d*_6_) δ = 12.08 (s, 1H), 8.18 (dd, *J* = 7.0, 1.4 Hz, 1H), 7.65–7.61 (m, 2H), 7.58–7.55 (m, 3H), 7.43 (dd, *J* = 7.0, 1.0 Hz, 1H), 7.21 (ddd, *J* = 9.1, 7.5, 1.3 Hz, 2H), 2.38 (t, *J* = 7.3 Hz, 2H), 1.48 (h, *J* = 7.4 Hz, 2H), 0.69 (t, *J* = 7.4 Hz, 3H); ^13^C NMR (125 MHz, DMSO-*d*_6_) δ = 197.2, 144.7, 135.9, 133.4, 130.3, 129.7, 128.9, 127.5, 123.2, 122.1, 122.0, 114.6, 112.1, 43.6, 18.5, 14.1. HRMS (ESI): *m*/*z* [M+H]^+^ calcd for C_18_H_18_NO: 264.1383; found: 264.1381.

*Phenyl(2-phenyl-1H-indol-3-yl)methanone (***3dd***)*. White solid; 88 mg, 59% yield, m.p. 231–232 °C (lit. 223–224 °C) [63]; ^1^H NMR (500 MHz, DMSO-*d*_6_) δ = 12.22 (s, 1H), 7.75 (d, *J* = 7.9 Hz, 1H), 7.54–7.50 (m, 3H), 7.40–7.37 (m, 2H), 7.37–7.34 (m, 1H), 7.27–7.23 (m, 4H), 7.23–7.19 (m, 2H), 7.18–7.14 (m, 1H); ^13^C NMR (125 MHz, DMSO-*d*_6_) δ = 192.6, 144.5, 140.3, 136.3, 132.0, 131.8, 130.0, 129.5, 128.9, 128.7, 128.5, 128.2, 123.3, 121.9, 121.0, 112.6, 112.3. HRMS (ESI): *m*/*z* [M+H]^+^ calcd for C_21_H_16_NO: 298.1226; found: 298.1224.

*1-(5-Bromo-1H-indol-3-yl)ethanone (***3ea***)*. White solid; 103 mg, 87% yield, m.p. 220–221 °C; ^1^H NMR (500 MHz, DMSO-*d*_6_) δ = 12.13 (s, 1H), 8.38 (d, *J* = 3.1 Hz, 1H), 8.31 (d, *J* = 2.0 Hz, 1H), 7.45 (d, *J* = 8.6 Hz, 1H), 7.34 (dd, *J* = 8.6, 2.0 Hz, 1H), 2.45 (s, 3H); ^13^C NMR (125 MHz, DMSO-*d*_6_) δ = 193.2, 136.0, 135.9, 127.5, 125.8, 123.9, 116.7, 114.9, 114.7, 27.6. HRMS (ESI): *m*/*z* [M+H]^+^ calcd for C_10_H_9_BrNO: 237.9862; found: 237.9860.

*1-(5-Bromo-1H-indol-3-yl)propan-1-one (***3eb***)*. White solid; 117 mg, 93% yield, m.p. 230–231 °C; ^1^H NMR (500 MHz, DMSO-*d*_6_) δ = 12.10 (s, 1H), 8.38 (d, *J* = 2.9 Hz, 1H), 8.33 (d, *J* = 1.9 Hz, 1H), 7.45 (d, *J* = 8.6 Hz, 1H), 7.34 (dd, *J* = 8.6, 2.0 Hz, 1H), 2.88 (q, *J* = 7.4 Hz, 2H), 1.10 (t, *J* = 7.4 Hz, 3H); ^13^C NMR (125 MHz, DMSO-*d*_6_) δ = 196.4, 135.8, 135.2, 127.7, 125.7, 123.9, 115.92, 114.9, 114.6, 32.3, 9.4. HRMS (ESI): *m*/*z* [M+H]^+^ calcd for C_11_H_11_BrNO: 22.0019; found: 252.0020.

*1-(5-Bromo-1H-indol-3-yl)butan-1-one (***3ec***)*. White solid; 111 mg, 84% yield, m.p. 232–233 °C (lit. 160–161 °C) [64]; ^1^H NMR (500 MHz, DMSO-*d*_6_) δ = 12.12 (s, 1H), 8.40 (d, *J* = 3.1 Hz, 1H), 8.34 (d, *J* = 2.0 Hz, 1H), 7.44 (d, *J* = 8.5 Hz, 1H), 7.34 (dd, *J* = 8.6, 2.0 Hz, 1H), 2.82 (t, *J* = 7.3 Hz, 2H), 1.66 (m, 2H), 0.93 (t, *J* = 7.4 Hz, 3H); ^13^CNMR (125 MHz, DMSO-*d*_6_) δ = 195.9, 135.8, 135.4, 127.6, 125.7, 124.0, 116.4, 114.9, 114.6, 41.1, 18.7, 14.3. HRMS (ESI): *m*/*z* [M+H]^+^ calcd for C_12_H_13_BrNO: 266.0715; found: 266.0720.

*(5-Bromo-1H-indol-3-yl)(phenyl)methanone (***3ed***)*. White solid; 105 mg, 79% yield, m.p. 272–273 °C (lit. 265–267 °C) [41]; ^1^H NMR (500 MHz, DMSO-*d*_6_) δ = 12.26 (s, 1H), 8.41 (s, 1H), 8.03 (s, 1H), 7.81 (d, *J* = 7.2 Hz, 2H), 7.63 (t, *J* = 7.3 Hz, 1H), 7.56 (t, *J* = 7.4 Hz, 2H), 7.51 (d, *J* = 8.6 Hz, 1H), 7.41 (dd, *J* = 8.6, 1.4 Hz, 1H); ^13^C NMR (125 MHz, DMSO-*d*_6_) δ = 190.3, 140.5, 137.3, 136.0, 131.8, 129.0, 128.9, 128.5, 126.2, 124.1, 115.2, 114.9, 114.8. HRMS (ESI): *m*/*z* [M+H]^+^ calcd for C_15_H_11_BrNO: 300.0019; found: 300.0021.

*3-Acetyl-1H-indole-5-carbonitrile (***3fa***)*. White solid; 86 mg, 93% yield, m.p. 271–272 °C (lit. 295–296 °C) [39]; ^1^H NMR (500 MHz, DMSO-*d*_6_) δ = 12.43 (s, 1H), 8.54 (s, 1H), 8.53 (d, *J* = 0.9 Hz, 1H), 7.66 (d, *J* = 8.4 Hz, 1H), 7.60 (dd, *J* = 8.4, 1.6 Hz, 1H), 2.49 (s, 3H); ^13^C NMR (125 MHz, DMSO-*d*_6_) δ = 193.3, 138.9, 136.9, 126.8, 126.1, 125.5, 120.6, 117.4, 114.0, 104.4, 27.7. HRMS (ESI): *m*/*z* [M+H]^+^ calcd for C_11_H_9_N_2_O: 185.0709; found: 185.0714.

*3-Propionyl-1H-indole-5-carbonitrile (***3fb***)*. White solid; 90 mg, 91% yield, m.p. 259–260 °C (lit. 252–254 °C) [39]; ^1^H NMR (500 MHz, DMSO-*d*_6_) δ = 12.41 (s, 1H), 8.55 (d, *J* = 0.9 Hz, 1H), 8.54 (s, 1H), 7.66 (d, *J* = 8.4 Hz, 1H), 7.59 (dd, *J* = 8.4, 1.6 Hz, 1H), 2.92 (q, *J* = 7.3 Hz, 2H), 1.12 (t, *J* = 7.4 Hz, 3H); ^13^C NMR (125 MHz, DMSO-*d*_6_) δ = 196.5, 138.9, 136.2, 126.8, 126.0, 125.6, 120.7, 116.6, 114.0, 104.3, 32.4, 9.2. HRMS (ESI): *m*/*z* [M+H]^+^ calcd for C_12_H_11_N_2_O: 199.0866; found: 199.0861.

3-Butyryl-1*H*-indole-5-carbonitrile (**3fc**). White solid; 93 mg, 88% yield, m.p. 209–210 °C; ^1^H NMR (500 MHz, DMSO-*d*_6_) δ = 12.42 (s, 1H), 8.56 (d, *J* = 2.0 Hz, 2H), 7.65 (d, *J* = 8.4 Hz, 1H), 7.59 (dd, *J* = 8.4, 1.5 Hz, 1H), 2.86 (t, *J* = 7.3 Hz, 2H), 1.71–1.63(m, 2H), 0.94 (t, *J* = 7.4 Hz, 3H); ^13^C NMR (125 MHz, DMSO-*d*_6_) δ = 196.1, 138.9, 136.5, 126.9, 126.1, 125.6, 120.7, 117.1, 114.1, 104.3, 41.2, 18.5, 14.3. HRMS (ESI): *m*/*z* [M+H]^+^ calcd for C_13_H_13_N_2_O: 213.1022; found: 213.1020.

*3-Benzoyl-1H-indole-5-carbonitrile (***3fd***)*. White solid; 65 mg, 53% yield, m.p. 202–204 °C; ^1^H NMR (500 MHz, DMSO-*d*_6_) δ = 12.56 (s, 1H), 8.63 (s, 1H), 8.20 (s, 1H), 7.84 (d, *J* = 6.8 Hz, 2H), 7.74–7.69 (m, 1H), 7.67–7.62 (m, 2H), 7.59–7.54 (m, 2H). ^13^C NMR (125 MHz, DMSO-*d*_6_) δ = 190.3, 140.2, 139.1, 138.4, 132.1, 129.0, 128.9, 126.0, 127.58, 126.6, 120.6, 115.6, 114.3, 104.6. HRMS (ESI): *m*/*z* [M+H]^+^ calcd for C_16_H_11_N_2_O: 247.0866; found: 247.0870.

*1-(5-Methyl-1H-indol-3-yl)ethanone (***3ga***)*. Pale yellow solid; 76 mg, 88% yield, m.p. 190–191 °C; ^1^H NMR (500 MHz, DMSO-*d*_6_) δ = 11.82 (s, 1H), 8.25 (d, *J* = 3.1 Hz, 1H), 7.99 (s, 1H), 7.35 (d, *J* = 8.2 Hz, 1H), 7.03 (dd, *J* = 8.2, 1.4 Hz, 1H), 2.43 (s, 3H), 2.40 (s, 3H); ^13^C NMR (125 MHz, DMSO-*d*_6_) δ = 193.0, 135.5, 134.8, 130.8, 126.0, 124.6, 121.5, 116.8, 112.2, 27.7, 21.8. HRMS (ESI): *m*/*z* [M+H]^+^ calcd for C_11_H_12_NO: 174.0913; found: 174.0920.

*1-(5-Methyl-1H-indol-3-yl)propan-1-one (***3gb***)*. Pale yellow solid; 78 mg, 83% yield, m.p. 225–226 °C; ^1^H NMR (500 MHz, DMSO-*d*_6_) δ = 11.78 (s, 1H), 8.25 (d, *J* = 3.1 Hz, 1H), 8.00 (s, 1H), 7.34 (d, *J* = 8.3 Hz, 1H), 7.02 (dd, *J* = 8.3, 1.4 Hz, 1H), 2.85 (q, *J* = 7.4 Hz, 2H), 2.40 (s, 3H), 1.11 (t, *J* = 7.4 Hz, 3H); ^13^C NMR (125 MHz, DMSO-*d*_6_) δ = 196.3, 135.4, 133.9, 130.8, 126.2, 124.6, 121.6, 116.1, 112.1, 32.3, 21.8, 9.7. HRMS (ESI): *m*/*z* [M+H]^+^ calcd for C_12_H_14_NO: 188.1070; found: 188.1076.

*1-(5-Methyl-1H-indol-3-yl)butan-1-one (***3gc***)*. Yellow solid; 76 mg, 76% yield, m.p. 195–196 °C; ^1^H NMR (500 MHz, DMSO-*d*_6_) δ = 11.79 (s, 1H), 8.26 (d, *J* = 3.1 Hz, 1H), 8.01 (s, 1H), 7.34 (d, *J* = 8.2 Hz, 1H), 7.02 (dd, *J* = 8.3, 1.2 Hz, 1H), 2.79 (t, *J* = 7.3 Hz, 2H), 2.40 (s, 3H), 1.70–1.62 (m, 2H), 0.93 (t, *J* = 7.4 Hz, 3H); ^13^CNMR (125 MHz, DMSO-*d*_6_) δ = 195.8, 135.4, 134.1, 130.8, 126.2, 124.6, 121.1, 116.6, 112.1, 41.1, 21.8, 18.9, 14.3. HRMS (ESI): *m*/*z* [M+H]^+^ calcd for C_13_H_16_NO: 202.1226; found: 202.1225.

*(5-Methyl-1H-indol-3-yl)(phenyl)methanone (***3gd***)*. White solid; 79 mg, 67% yield, m.p. 227–228 °C (lit. 228 °C) [56]; ^1^H NMR (500 MHz, DMSO-*d*_6_) δ = 11.97 (s, 1H), 8.09 (s, 1H), 7.87 (d, *J* = 1.6 Hz, 1H), 7.79–7.78 (m, 1H), 7.77–7.76 (m, 1H), 7.62–7.58 (m, 1H), 7.56–7.52 (m, 2H), 7.41 (d, *J* = 8.3 Hz, 1H), 7.10 (dd, *J* = 8.3, 1.5 Hz, 1H), 2.45 (s, 3H); ^13^C NMR (125 MHz, DMSO-*d*_6_) δ = 190.4, 141.1, 136.2, 135.5, 131.4, 131.2, 128.8, 128.8, 127.0, 125.1, 121.7, 115.1, 112.3, 21.8. HRMS (ESI): *m*/*z* [M+H]^+^ calcd for C_16_H_14_NO: 236.1070; found: 236.1074.

*1-(5-Methoxy-1H-indol-3-yl)ethanone (***3ha***)*. Pale yellow solid; 76 mg, 80% yield, m.p. 209–210 °C (lit. 170 °C) [65]; ^1^H NMR (500 MHz, DMSO-*d*_6_) δ = 11.81 (s, 1H), 8.25 (d, *J* = 3.1 Hz, 1H), 7.68 (d, *J* = 2.5 Hz, 1H), 7.36 (d, *J* = 8.8 Hz, 1H), 6.84 (dd, *J* = 8.8, 2.6 Hz, 1H), 3.77 (s, 3H), 2.43 (s, 3H); ^13^C NMR (125 MHz, DMSO-*d*_6_) *δ* 193.0, 155.8, 135.0, 132.0, 126.5, 117.1, 113.2, 113.0, 103.5, 55.7, 27.6. HRMS (ESI): *m*/*z* [M+H]^+^ calcd for C_11_H_12_NO_2_: 190.0863; found: 190.0860.

*1-(5-Methoxy-1H-indol-3-yl)propan-1-one (***3hb***)*. Yellow solid; 76 mg, 75% yield, m.p. 202–203 °C; ^1^H NMR (500 MHz, DMSO-*d*_6_) δ = 11.78 (s, 1H), 8.25 (d, *J* = 3.2 Hz, 1H), 7.71 (d, *J* = 2.5 Hz, 1H), 7.35 (d, *J* = 8.8 Hz, 1H), 6.83 (dd, *J* = 8.8, 2.5 Hz, 1H), 3.78 (s, 3H), 2.85 (q, *J* = 7.4 Hz, 2H), 1.11 (t, *J* = 7.4 Hz, 3H); ^13^C NMR (125 MHz, DMSO-*d*_6_) δ = 196.3, 155.8, 134.1, 131.9, 126.7, 116.3, 113.2, 113.0, 103.6, 55.7, 32.2, 9.6. HRMS (ESI): *m*/*z* [M+H]^+^ calcd for C_12_H_14_NO_2_: 204.1019; found: 204.1022.

*1-(5-Methoxy-1H-indol-3-yl)butan-1-one (***3hc***)*. Yellow solid; 74 mg, 68% yield, m.p. 151–152 °C; ^1^H NMR (500 MHz, DMSO-*d*_6_) δ = 11.80 (s, 1H), 8.26 (d, *J* = 3.1 Hz, 1H), 7.72 (d, *J* = 2.6 Hz, 1H), 7.35 (d, *J* = 8.8 Hz, 1H), 6.83 (dd, *J* = 8.8, 2.6 Hz, 1H), 3.78 (s, 3H), 2.79 (t, *J* = 7.3 Hz, 2H), 1.69–1.63 (m, 2H), 0.94 (t, *J* = 7.4 Hz, 3H); ^13^C NMR (125 MHz, DMSO-*d*_6_) δ = 195.8, 155.8, 134.3, 131.9, 126.6, 116.8, 113.2, 113.1, 103.5, 55.7, 41.1, 18.8, 14.4. HRMS (ESI): *m*/*z* [M+H]^+^ calcd for C_13_H_16_NO_2_: 218.1176; found: 218.1180.

*(5-Methoxy-1H-indol-3-yl)(phenyl)methanone (***3hd***)*. Pale yellow solid; 80 mg, 64% yield, m.p. 213–215 °C (lit. 169–170 °C) [66]; ^1^H NMR (500 MHz, DMSO-*d*_6_) δ = 11.97 (s, 1H), 7.87 (d, *J* = 3.2 Hz, 1H), 7.79 (d, *J* = 2.5 Hz, 1H), 7.78 (d, *J* = 1.0 Hz, 1H), 7.77–7.62 (m, 1H), 7.62–7.58 (m, 1H), 7.57–7.52 (m, 2H), 7.42 (d, *J* = 8.8 Hz, 1H), 6.90 (dd, *J* = 8.8, 2.6 Hz, 1H), 3.81 (s, 3H); ^13^C NMR (125 MHz, DMSO-*d*_6_) δ = 190.4, 156.0, 141.1, 136.4, 132.0, 131.4, 128.9, 128.8, 127.5, 115.3, 113.5, 113.5, 103.7, 55.8. HRMS (ESI): *m*/*z* [M+H]^+^ calcd for C_16_H_14_NO_2_: 252.1019; found: 252.1020.

*1-(6-Fluoro-1H-indol-3-yl)ethanone (***3ia***)*. Pale yellow solid; 81 mg, 92% yield, m.p. 227–228 °C (lit. 236 °C) [67]; ^1^H NMR (500 MHz, DMSO-*d*_6_) δ = 11.98 (s, 1H), 8.33 (d, *J* = 3.0 Hz, 1H), 8.15 (dd, *J* = 8.7, 5.7 Hz, 1H), 7.26 (dd, *J* = 9.7, 2.4 Hz, 1H), 7.06–7.02 (m, 1H), 2.44 (s, 3H); ^13^C NMR (125 MHz, DMSO-*d*_6_) δ = 193.09, 159.68 (d, *J* = 235.0 Hz), 137.16 (d, *J* = 12.5 Hz), 135.5, 122.9 (d, *J* = 10.0 Hz), 122.4, 117.2, 110.4 (d, *J* = 23.8 Hz), 98.8 (d, *J* = 25.0 Hz), 27.6. HRMS (ESI): *m*/*z* [M+H]^+^ calcd for C_10_H_9_FNO: 178.0663; found: 178.0666.

*1-(6-Fluoro-1H-indol-3-yl)propan-1-one (***3ib***)*. Pale yellow solid; 84 mg, 88% yield, m.p. 215–216 °C; ^1^H NMR (500 MHz, DMSO-*d*_6_) δ = 11.95 (s, 1H), 8.33 (d, *J* = 3.0 Hz, 1H), 8.17 (dd, *J* = 8.7, 5.7 Hz, 1H), 7.25 (dd, *J* = 9.7, 2.3 Hz, 1H), 7.07–7.01 (m, 1H), 2.87 (q, *J* = 7.4 Hz, 2H), 1.11 (t, *J* = 7.4 Hz, 3H); ^13^C NMR (125 MHz, DMSO-*d*_6_) δ = 196.3, 159.7 (d, *J* = 235.0 Hz), 137.1 (d, *J* = 12.5 Hz), 134.6 (d, *J* = 1.3 Hz), 122.9 (d, *J* = 10.0 Hz), 122.6, 116.4, 110.3, (d, *J* = 23.7 Hz), 98.7 (d, *J =* 25.0 Hz), 32.2, 9.5. HRMS (ESI): *m*/*z* [M+H]^+^ calcd for C_11_H_11_FNO: 192.0819; found: 192.0823.

*1-(6-Fluoro-1H-indol-3-yl)butan-1-one (***3ic***)*. White solid; 83 mg, 81% yield, m.p. 190–191 °C; ^1^H NMR (500 MHz, DMSO-*d*_6_) δ = 11.97 (s, 1H), 8.34 (s, 1H), 8.17 (dd, *J* = 8.7, 5.7 Hz, 1H), 7.25 (dd, *J* = 9.7, 2.3 Hz, 1H), 7.06–7.01 (m, 1H), 2.81 (t, *J* = 7.3 Hz, 2H), 1.71–1.59 (m, 2H), 0.93 (t, *J* = 7.4 Hz, 3H); ^13^C NMR (125 MHz, DMSO-*d*_6_) δ = 195.8, 159.7 (d, *J* = 235.0 Hz), 137.2 (d, *J* = 12.5 Hz), 134.8 (d, *J* = 1.3 Hz), 123.0 (d, *J* = 10.0 Hz), 122.6, 116.9, 110.3 (d, *J* = 23.8 Hz), 98.7 (d, *J* = 26.3 Hz), 41.0, 18.7, 14.3. HRMS (ESI): *m*/*z* [M+H]^+^ calcd for C_12_H_13_FNO: 206.0976; found: 206.0972.

*(6-Fluoro-1H-indol-3-yl)(phenyl)methanone (***3id***)*. White solid; 80 mg, 67% yield, m.p. 268–269 °C; ^1^H NMR (500 MHz, DMSO-*d*_6_) δ = 12.12 (s, 1H), 8.25 (dd, *J* = 8.7, 5.7 Hz, 1H), 7.96 (d, *J* = 2.7 Hz, 1H), 7.83–7.77 (m, 2H), 7.64–7.60 (m, 1H), 7.55 (t, *J* = 7.4 Hz, 2H), 7.32 (dd, *J* = 9.6, 2.3 Hz, 1H), 7.14–7.09 (m, 1H); ^13^C NMR (125 MHz, DMSO-*d*_6_) δ = 190.3, 159.9 (d, *J* = 236.3 Hz), 140.7, 1373 (d, *J* = 12.5 Hz), 136.9, 131.6, 128.9 (d, *J* = 3.8 Hz), 123.1 (d, *J* = 10.0 Hz), 123.1, 115.4, 110.7 (d, *J* = 23.8 Hz), 99.0 (d, *J* = 26.3 Hz). HRMS (ESI): *m*/*z* [M+H]^+^ calcd for C_15_H_11_FNO: 240.0819; found: 240.0825.

*1-(7-Bromo-1H-indol-3-yl)ethanone (***3ja***)*. White solid; 108 mg, 91% yield, m.p. 184–185 °C (lit. 190–191 °C) [67]; ^1^H NMR (500 MHz, DMSO-*d*_6_) δ = 12.19 (s, 1H), 8.37 (s, 1H), 8.19 (dd, *J* = 7.9, 0.8 Hz, 1H), 7.45 (dd, *J* = 7.6, 0.8 Hz, 1H), 7.13 (t, *J* = 7.8 Hz, 1H), 2.48 (s, 3H); ^13^C NMR (125 MHz, DMSO-*d*_6_) δ = 193.4, 135.6, 135.5, 127.4, 125.9, 123.6, 121.2, 118.1, 105.1, 27.9. HRMS (ESI): *m*/*z* [M+H]^+^ calcd for C_10_H_9_BrNO: 237.9862; found: 237.9868.

*1-(7-Bromo-1H-indol-3-yl)propan-1-one (***3jb***)*. White solid; 112 mg, 89% yield, m.p. 148–149 °C; ^1^H NMR (500 MHz, DMSO-*d*_6_) δ = 12.16 (s, 1H), 8.36 (d, *J* = 3.1 Hz, 1H), 8.21 (d, *J* = 7.5 Hz, 1H), 7.44 (dd, *J* = 7.6, 0.9 Hz, 1H), 7.13 (t, *J* = 7.8 Hz, 1H), 2.92 (q, *J* = 7.4 Hz, 2H), 1.11 (t, *J* = 7.4 Hz, 3H); ^13^C NMR (125 MHz, DMSO-*d*_6_) δ = 196.6, 135.5, 134.8, 127.6, 125.8, 123.6, 121.3, 117.4, 105.1, 32.4, 9.3. HRMS (ESI): *m*/*z* [M+H]^+^ calcd for C_11_H_11_BrNO: 22.0019; found: 252.0016.

*1-(7-Bromo-1H-indol-3-yl)butan-1-one (***3jc***)*. White solid; 117 mg, 88% yield, m.p. 158–159 °C; ^1^H NMR (500 MHz, DMSO-*d*_6_) δ = 12.17 (s, 1H), 8.38 (s, 1H), 8.21 (d, *J* = 7.3 Hz, 1H), 7.44 (d, *J* = 7.6 Hz, 1H), 7.13 (t, *J* = 7.8 Hz, 1H), 2.86 (t, *J* = 7.3 Hz, 2H), 1.70–1.62 (m, 2H), 0.94 (t, *J* = 7.4 Hz, 3H); ^13^C NMR (125 MHz, DMSO-*d*_6_) δ = 196.2, 135.5, 135.0, 127.6, 125.9, 123.6, 121.3, 117.8, 105.1, 41.2, 18.6, 14.3. HRMS (ESI): *m*/*z* [M+H]^+^ calcd for C_12_H_13_BrNO: 266.0715; found: 266.0717.

*(7-Bromo-1H-indol-3-yl)(phenyl)methanone (***3jd***)*. White solid; 123mg, 82% yield, m.p. 204–205 °C; ^1^H NMR (500 MHz, DMSO-*d*_6_) δ = 12.38 (s, 1H), 8.27 (d, *J* = 7.9 Hz, 1H), 7.90 (s, 1H), 7.82 (s, 1H), 7.81 (d, *J* = 1.4 Hz, 1H), 7.66–7.62 (m, 1H), 7.59–7.54 (m, 2H), 7.52 (d, *J* = 7.6 Hz, 1H), 7.21 (t, *J* = 7.8 Hz, 1H); ^13^C NMR (125 MHz, DMSO-*d*_6_) δ = 190.5, 140.5, 136.6, 135.6, 131.9, 129.0, 128.9, 128.4, 126.3, 123.9, 121.4, 116.4, 105.2. HRMS (ESI): *m*/*z* [M+H]^+^ calcd for C_15_H_11_BrNO: 300.0019; found: 300.0018.

### 3.2. Up to Ten Scale Synthesis of Selected (***3aa***)

A total of 11.7 g (0.1 mol) of indole **1** was dissolved in 200 mL of DCM and 12.6 g (12.0 mL, 0.12 mol) of acetic anhydride **2a** was added. Then, 14.6 g (0.1 mol) of BF_3_^.^Et_2_O was added dropwise to the stirred mixture at room temperature. After finishing the addition, the reaction mixture was stirred continuously at room temperature until completed. Then, 100 mL of saturated sodium bicarbonate was added and stirred at room temperature for about 0.5 h. The organic layer was separated and the water phase was extracted with DCM (2 × 100 mL). The organic layer was combined, washed with saturated sodium bicarbonate (2 × 100 mL) and dried over Na_2_SO_4_. The solvent was removed and the residue was purified by column chromatography on silica gel or recrystallized from MeOH/H_2_O (5:1) to give **3aa** 12.7 g in 80% yield.

### 3.3. Procedure for Synthesis of ***4***

A solution of synthesized **3aa** (10.0 mmol, 1.59 g), NH_2_OH·HCl (20.0 mmol, 1.39 g) and pyridine (30.0 mmol, 2.4 mL) in MeOH (30 mL) was stirred at room temperature for about 18~24 h. The reaction mixture was evaporated to remove MeOH in vacuo, and to the residue was then added to water (50 mL). After extraction with DCM (2 × 50 mL), the combined organic layers were washed with brine, dried over Na_2_SO_4_, and filtered. Volatiles were removed under vacuum to give the oxime **4** as a white solid without any purification to the next step (1.2 g, 69% yield, m.p. 144–146 °C (lit. 147–148 °C)) [68].

### 3.4. Procedure for Synthesis of ***5***

A Schlenk tube was charged with **4** (0.5 mmol, 87 mg), and DCE (5 mL). *t*-BuOK (0.75 mmol, 1.5 equiv) was added in one portion at room temperature under a nitrogen atmosphere. The mixture was stirred at room temperature for 5 min. Then, Ph_2_IOTf (0.75 mmol, 220 mg, 1.5 equiv) was added in one portion. The reaction was stirred at room temperature for 4 h. At this time, the DCE was removed under reduced pressure, and the crude product was purified by column chromatography on silica gel using petroleum ether/ethyl acetate 1/6 to 1/3 to provide product **5**. 

(*E*)-1-(1*H*-indol-3-yl)ethanone *O*-phenyl oxime (**5**). Yellow oil; 106 mg, yield 85%; ^1^H NMR (400 MHz, CDCl_3_) δ = 8.50–8.46 (m, 1H), 8.27 (s, 1H), 7.43 (d, J = 2.8 Hz, 1H), 7.42 (s, 2H), 7.41 (s, 2H), 7.37–7.34 (m, 1H), 7.32–7.28 (m, 2H), 7.10–7.04 (m, 1H), 2.47 (s, 3H); ^13^C NMR (100 MHz, CDCl3) δ = 159.9, 155.4, 137.5, 136.8, 130.3, 129.4, 127.5, 126.2, 124.5, 123.4, 121.7, 121.5, 114.7, 113.9, 111.2, 13.6; HRMS (ESI): *m*/*z* [M+H]^+^ calcd for C_16_H_15_N_2_O: 251.1179; found: 251.1171.

### 3.5. Procedure for Synthesis of ***6***

A Schlenk tube, open to air, was charged with **5** (0.5 mmol, 125 mg) and 1,4-dioxane (5 mL). A total of 4 M HCl (0.75 mL, 6 equiv) and H_2_O (0.054 mL, 6 equiv) was added in one portion at room temperature. The mixture was stirred at 80 °C. The reaction was monitored by TLC until **5** was consumed completely (8−12 h). At this time, the solvent was removed under reduced pressure, and the residue was washed with saturated sodium bicarbonate (10 mL). Then, after extraction with DCM (3 × 10 mL), the combined organic layers were dried over Na_2_SO_4_ and filtered. DCM was removed under reduced pressure, and the crude product was purified by column chromatography on silica gel using petroleum ether/ethyl acetate 1/10 to 1/8 to provide product **6**.

3-(Benzofuran-2-yl)-1H-indole (**6**). White solid; 72 mg, 62% yield, m.p. 147–148 °C (lit. 162–163 °C) [69]; ^1^H NMR (500 MHz, CDCl_3_) δ = 8.33 (s, 1H), 8.14–8.08 (m, 1H), 7.71 (s, 1H), 7.63 (dd, *J* = 5.9, 2.3 Hz, 1H), 7.56 (d, *J* = 6.9 Hz, 1H), 7.45–7.41 (m, 1H), 7.36–7.31 (m, 2H), 7.31–7.27 (m, 2H), 6.97 (s, 1H); ^13^C NMR (125 MHz, CDCl_3_) δ = 154.0, 153.0, 136.6, 129.9, 124.6, 123.3, 123.15, 123.06, 122.8, 121.1, 120.34, 120.29, 111.7, 110.8, 108.7, 99.7; HRMS (ESI): *m*/*z* [M+H]^+^ calcd for C_16_H_12_NO: 234.0913; found: 234.0916.

## 4. Conclusions

In conclusion, we have developed a mild and efficient synthetic method for the BF_3_·Et_2_O-promoted acylation of free (N-H) indoles with anhydrides. This protocol afforded a variety of 3-acylindoles in good-to-excellent yields with high regioselectivity and was easily up to 10 g scale. 3-Benzofuran-2-yl indole can be synthesized in good yield in three steps. This protocol accomplished the challenging acylation of free (N-H) indoles successfully. Further studies on their synthetic applications are currently underway in our laboratory.

## Data Availability

The data presented in this study are available in the Appendix A.

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
