# Peer review of "Boron Trifluoride Etherate Promoted Regioselective 3-Acylation of Indoles with Anhydrides"

_molecules, 2022, doi:10.3390/molecules27238281_

Round 1

Reviewer 2 Report

The authors developed a practical synthetic method for 3-acylated indoles. The protecting group-free acylation, which is convenient in organic syntheses, was achieved. The authors emphasized that the novelties and advantages of this work are that protecting group-free strategy and practical utility of the BF3OEt2  complex compared to other conventional Lewis acids. These sound attractive but should be carefully examined. 

Comments:

1) The results using the other Lewis acids under the optimal conditions in Table 1 or SI should be described in order to prove the advantage of the use of BF3OEt2 for the protecting group-free acylation.

2) BF3OEt2 was commented to be the more moisture tolerant, air-stable and easy-to-handle, but it may not be common sense. According to a few chemical catalogues, it is regarded to be moisture sensitive, highly flammable, and needed to be stored at lower temperature. It is difficult to say the relative dangers or utility, BF3OEt2 is also needed to be carefully used as well. Furthermore, BF3OEt2 may form an explosive peroxide in contact with air or oxygen. The presented reaction should be conducted under nitrogen or argon atmosphere from the viewpoint of safety issue, although the procedure described "performed under air conditions".  

3) 13C NMR chart of the compound 6 is dirty, suggesting a significant error of its yield.

Round 2

Reviewer 2 Report

The manuscript was revised in accordance with my requests, and I think it can be accepted for publication.

Author Response

Thank you very much for your review and agreeing to accept manuscript for publication!